# Poisson-Induced Potentials for Contractive representations

**Guillermo Moreno​Department of Electrical Engineering**
Pontifical Catholic University of Rio de Janeiro
Gávea, Rio de Janeiro - State of Rio de Janeiro, Brasil
`guillermo@ica.ele.puc-rio.br`

## Abstract

Contractive Auto-Encoders (CAE) finds orthogonal representations by penalizing the Frobenius norm of the encoder Jacobian. This work provides a Poisson-based reformulation of the contractive penalty that yields a geometric decomposition of the regularizer. By introducing an auxiliary potential field $v_\phi$ defined as the solution of a Poisson equation whose source is the contractive term, and applying Green's first identity. The expected contractive penalty can be expressed as the sum of a boundary-flux contribution and an interior score–potential coupling term. The latter recovers known connections between regularized autoencoders and the score of the data distribution, while the boundary-flux term motivates an additional mechanism: probing the effective support of the data through out-of-distribution transformations. Inspired by diffusion models, we approximate the boundary term using a corruption operator, which induces both the evaluation points and a normal-like direction. We validate the proposed viewpoint on toy datasets and image data, visualizing $f(x)$, $\|\nabla_x f(x)\|$, the induced potential $v(x)$, and $\nabla v(x)$ under varying perturbation strengths, and we observe that the Poisson potential provides a global summary of contractivity that is sensitive to corruption-driven departures from the data manifold.

## 1 Introduction

Autoencoder architectures are primarily motivated by representation learning: they map inputs to intermediate codes that are constrained to capture structure beyond trivial identity mapping. This is typically achieved by imposing inductive biases or explicit penalties on intermediate representations, as in Contractive Auto-Encoders (CAE) Rifai et al. (2011) and Variational Autoencoders (VAE) Kingma & Welling (2022). In CAE, the encoder is encouraged to be locally insensitive to input perturbations by penalizing the Frobenius norm of the encoder Jacobian, promoting stable representations and emphasizing directions aligned with the data manifold. In VAE-based models, a probabilistic bottleneck is enforced by matching an approximate posterior to a prior, motivating a rich family of variants that alter the latent prior, posterior, or hierarchical structure Dilokthanakul et al. (2017); Tomczak & Welling (2018); Sønderby et al. (2016); van den Oord et al. (2018). While these approaches yield useful representations and explicit likelihood-based modeling, a persistent challenge for explicit latent-variable autoencoders is limited generative fidelity, including phenomena such as posterior collapse and mode dropping. Numerous works attempt to mitigate these limitations by increasing latent expressiveness or coupling autoencoders with more flexible generative mechanisms, such as normalizing flows or energy-based components Papamakarios et al. (2021); Xiao et al. (2021). Despite these advances, autoencoder-based explicit models often remain behind leading implicit generative models in sample quality.

From a broader perspective, autoencoder models belong to the class of explicit generative models, they are widely used when interpretable or transferable representations are required. For instance, VAE-style models have become particularly popular in scientific applications such as genomics Ashuach et al. (2023); Chandrashekar et al. (2025). In contrast, the strongest generative performance on high-dimensional perceptual data is typically achieved by implicit modeling approaches, including diffusion models Sohl-Dickstein et al. (2015); Ho et al. (2020), generative adversarial networks Goodfellow et al. (2014), and more recently flow matching Lipman et al. (2023). Under-

standing the principles behind the effectiveness of such implicit approaches is a motivation of this work.

A recurring motif across generative learning is the deliberate use of "negative" or out-of-distribution information. Contrastive Divergence (CD) rey E. Hinton (2000) exemplifies this idea by updating parameters using both observed samples and samples obtained from short-run Markov chains, thereby contrasting data likelihood with model-generated "nearby" negatives. Denoising autoencoders also instantiate this motif: they are trained to reconstruct clean data from corrupted observations, and can be interpreted as learning local properties of the data distribution through corruption and reconstruction Bengio et al. (2013). Diffusion-based models generalize this principle by training networks to invert a controlled corruption process across a range of noise levels, which ultimately yields high-quality generative sampling Sohl-Dickstein et al. (2015); Ho et al. (2020).

Related principles have also influenced self-supervised representation learning, where invariances are induced by comparing transformed views of the same sample. Methods such as SimCLR Chen et al. (2020) learn representations by contrasting pairs of augmentations passed through shared encoders, while masked modeling approaches such as MAE He et al. (2021) learn robust representations by reconstructing inputs from strong structured perturbations. Teacher-student approaches such as DINO Caron et al. (2021) combine multi-view learning with momentum-based target networks He et al. (2020), showing that strong representation learning can emerge from consistency constraints under perturbations.

We reformulate the CAE contractive penalty by introducing a Poisson potential whose source is the encoder Jacobian norm. Green's first identity yields a decomposition into a *boundary flux* term and an interior *score–potential coupling* term, extending known links between regularized autoencoders and the data score Alain & Bengio (2014). Motivated by denoising and diffusion training, we treat the boundary contribution through explicit perturbations that generate corrupted samples, producing a regularizer applicable to CAE-style reconstruction and other downstream objectives.

## 2 PROBLEM SETUP

Let $\Omega \subset \mathbb{R}^d$ denote the ambient input domain and let $p_D$ be the (unknown) data distribution supported on (a subset of) $\Omega$. We consider an autoencoder composed of an encoder and decoder,

$$f_\phi : \Omega \to \mathbb{R}^m, \qquad g_\theta : \mathbb{R}^m \to \Omega, \tag{1}$$

where $f_\phi(x)$ produces a representation (hidden state) and $g_\theta(f_\phi(x))$ reconstructs the input. The downstream objective is generic; in the contractive autoencoder (CAE) case it is a reconstruction loss. In the experiments we use

$$\mathcal{L}_{\text{rec}}(\theta, \phi) = \mathbb{E}_{x \sim p_D} \left[ \ell(g_\theta(f_\phi(\tilde{x})), x) \right] \tag{2}$$

**Contractive regularization.** The classical CAE penalizes the Frobenius norm of the Jacobian of the encoder representation with respect to the input:

$$\mathcal{R}_{\text{CAE}}(\phi) = \mathbb{E}_{x \sim p_D} \left[ \|J_{f_\phi}(x)\|_F^2 \right], \quad J_{f_\phi}(x) = \nabla_x f_\phi(x) \in \mathbb{R}^{m \times d}, \quad s_\phi(x) := \|J_{f_\phi}(x)\|_F^2 \tag{3}$$

## 3 POISSON REFORMULATION OF THE CONTRACTIVE REGULARIZER

This section introduces a Poisson potential associated with contractivity and derives a decomposition of $\mathcal{R}_{\text{CAE}}(\phi)$ using Green's first identity. A complete derivation with technical conditions is provided in Appendix B.

### 3.1 POISSON POTENTIAL INDUCED BY CONTRACTIVITY

Define the scalar source $s_\phi(x)$ as in equation 3. We introduce an auxiliary scalar field $v_\phi : \Omega \to \mathbb{R}$ as the (weak) solution of the Poisson problem

$$\Delta v_\phi(x) = s_\phi(x) \quad \text{in } \Omega, \tag{4}$$

supplemented with boundary conditions. In the simplest case we assume homogeneous Dirichlet boundary conditions $v_\phi|_{\partial\Omega} = 0$ (other choices are discussed in Appendix B).

Formally, the solution admits a Green's function representation

$$v_\phi(x) = \int_\Omega G_\Omega(x, y) \, s_\phi(y) \, dy, \tag{5}$$

where $G_\Omega$ is the Green's function of the Laplacian on $\Omega$ with the chosen boundary conditions. In practice, we approximate equation 5 by a Monte Carlo quadrature over samples (Section 5).

### 3.2 GREEN DECOMPOSITION: BOUNDARY FLUX AND SCORE COUPLING

Let $u = p_D$ and $v = v_\phi$ in Green's first identity:

$$\int_\Omega u \, \Delta v \, dx = \int_{\partial\Omega} u \, \partial_n v \, dS - \int_\Omega \nabla u \cdot \nabla v \, dx, \tag{6}$$

where $\partial_n v = \nabla v \cdot n$ is the outward normal derivative and $dS$ is the surface measure.

Using $\Delta v_\phi = s_\phi$ and $\nabla p_D = p_D \, \nabla \log p_D$, we obtain the decomposition

$$\mathcal{R}_{\text{CAE}}(\phi) = \int_\Omega p_D(x) \, s_\phi(x) \, dx \; = \; \underbrace{\int_{\partial\Omega} p_D(s) \, \partial_n v_\phi(s) \, dS(s)}_{\text{boundary flux}} - \underbrace{\int_\Omega \nabla p_D(x) \cdot \nabla v_\phi(x) \, dx}_{\text{score–potential coupling}} \, .$$

Equation equation 7 reveals that contractive regularization can be interpreted through a flux term plus an interior term coupling $v_\phi$ with the unknown score of the data distribution.

### 3.3 PROJECTION / CORRUPTION OPERATOR VIEW OF THE BOUNDARY TERM

In high-dimensional data, the geometric boundary $\partial\Omega$ is not directly accessible. Motivated by modern generative modeling practice (denoising, diffusion), we interpret the perturbation operator $\Pi_\psi$ as a *boundary-reaching* mapping: it transforms in-distribution inputs into corrupted / out-of-distribution samples. This provides a tractable surrogate for evaluating the boundary flux.

We propose to model the boundary flux term via an expectation over $x \sim p_D$ by using the same corruption operator to define both: (i) the evaluation point $s = \Pi(x)$, and (ii) a normal-like direction induced by the corruption displacement. Concretely, we define and approximate the boundary flux contribution by the directional derivative at the corrupted point,

$$\int_{\partial\Omega} p_D(s) \, \partial_n v_\phi(s) \, dS(s) \; \approx \; \mathbb{E}_{x \sim p_D}\Big[\nabla v_\phi(\Pi(x)) \cdot n(x)\Big], \quad n(x) := \frac{\Pi(x) - x}{\|\Pi(x) - x\| + \varepsilon} \tag{7}$$

This is a modeling choice: equation 7 replaces an intractable surface integral by a corruption-driven flux proxy, aligning the method with the core motif of corruption-based generative learning.

### 3.4 FINAL TRAINING OBJECTIVE

Combining reconstruction equation 2 with the Poisson-induced regularization gives the full objective

$$\min_{\theta, \phi} \; \mathcal{L}_{\text{rec}}(\theta, \phi) \; + \; \lambda \, \mathcal{L}_{\text{Pois}}(\phi), \tag{8}$$

where the proposed Poisson regularizer is

$$\mathcal{L}_{\text{Pois}}(\phi) := \mathbb{E}_{x \sim p_D}\Big[\nabla v_\phi(\Pi(x)) \cdot n(x)\Big] - \mathbb{E}_{x \sim p_D}\Big[\nabla \log p_D(x) \cdot \nabla v_\phi(x)\Big]. \tag{9}$$

Finally, $v_\phi$ depends on $f_\phi$ through the source $s_\phi(x) = \|J_{f_\phi}(x)\|_F^2$ and the Poisson equation equation 4. In our implementation, we approximate $v_\phi$ and $\nabla v_\phi$ via sample-based quadrature of the Green representation equation 5.

3.5 ESTIMATING $s_\phi(x)$, $v_\phi(x)$, AND $\nabla v_\phi(x)$

The Poisson potential is approximated by Monte Carlo quadrature of equation 5, with $z_i = f_\phi(x_i)$:

$$v_\phi(x_i) \approx \frac{1}{M} \sum_{j=1}^{M} G_\varepsilon(z_i, z_j)\, s_\phi(x_j), \quad \nabla v_\phi(x) \approx \frac{1}{M} \sum_{j=1}^{M} \nabla G_\varepsilon(z_i, z_j)\, s_\phi(x_j). \qquad (10)$$

## 4 INTERPRETATION AND CONNECTIONS

From an intuitive perspective, the decomposed loss equation 9 contains a standard data term together with a regularizer that can be read as a *corruption-driven flux* plus an *interior coupling to the data score*. This structure aligns with several motifs in the generative modeling literature:

**Denoising and diffusion.** Equation equation 7 uses perturbed samples $\Pi(x)$ and directional derivatives along the corruption displacement, conceptually resembling denoising objectives that learn from corrupted data. The coupling term in equation 9 involves $\nabla \log p_D$, reminiscent of score-based formulations; extending the method by explicitly estimating the score would further strengthen this connection.

**Contrastive objectives.** The use of out-of-distribution (corrupted) samples echoes contrastive divergence style training where learning is informed by both data and samples produced by a transformation away from the data manifold. Here, the transformation $\Pi(\cdot)$ is explicit and the penalty is expressed through a potential induced by encoder contractivity.

**Representation learning.** Contractive penalties bias representations to be stable under perturbations. The proposed Poisson potential makes this bias explicit as a field $v_\phi$ that summarizes contractive intensity across the space and produces gradients that can be used to regularize how representations behave under corruption.

## 5 EXPERIMENTS

**Datasets and architectures.** We consider a collection of toy datasets to probe the behavior of the proposed regularizer under controlled geometric structure. The model is a neural network autoencoder with encoder $f_\phi$ and decoder $g_\theta$. Unless stated otherwise, the corruption operator is additive Gaussian noise. Also, we evaluate other perturbation operators that preserve global content while altering local geometry, including random rotations and random zoom (scaling) transformations. These operators are used both to generate corrupted inputs for denoising-style training and to probe out-of-distribution behavior of the learned representations.

**Visualization protocol.** We visualize, for batches of samples: (i) $f_\phi(x)$ via 2D PCA projections of $z = f_\phi(x)$; (ii) $s_\phi(x)$ histograms (contractivity intensity); (iii) $v_\phi(x)$ histograms; and (iv) $\|\nabla v_\phi(z)\|$ histograms. We compare these quantities for clean inputs and corrupted inputs $\tilde{x} = \Pi_\psi(x)$.

## 6 DISCUSSION

The Poisson reformulation emphasizes that contractive regularization is not merely a local smoothness constraint: through equation 4–equation 5, it induces a *global potential field* $v_\phi$ whose gradients can be probed along corruption trajectories. The decomposition equation 9 suggests two complementary effects: a boundary-driven flux term that is naturally aligned with corruption-based training, and an interior score–potential coupling that links contractivity to the geometry of the data distribution, this result has been already shown in Alain & Bengio (2014), but this derivation offers a theoretical venue to futher interpret this result and connection with corruption-based methods.

A practical implication is that one can design regularizers that explicitly depend on corrupted / out-of-distribution inputs, which is consistent with the broader theme of modern implicit generative learning. Futher work can be directed towards exploring controlled boundary to define out-of-distribution or pertenence boundaries. Another interesting path is observing the implication of a

more sofisticated perturbation mapping, either parametrized or completely stochastic. different perturbation processes may induce different geometric biases, similarly to the role of noise schedules in diffusion models.

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

## A  COMPLETE DERIVATION OF THE GREEN DECOMPOSITION

We provide the full derivation of equation 7. Let $p_D$ be sufficiently smooth on $\Omega$ and assume $v_\phi$ solves equation 4 with boundary conditions such that Green's first identity applies. Starting from

$$\mathcal{R}(\phi) = \int_\Omega p_D(x)\, s_\phi(x)\, dx = \int_\Omega p_D(x)\, \Delta v_\phi(x)\, dx, \tag{11}$$

apply Green's first identity equation 6 with $u = p_D$ and $v = v_\phi$ to obtain

$$\int_\Omega p_D\, \Delta v_\phi\, dx = \int_{\partial\Omega} p_D\, \partial_n v_\phi\, dS - \int_\Omega \nabla p_D \cdot \nabla v_\phi\, dx. \tag{12}$$

Finally, using $\nabla p_D = p_D \nabla \log p_D$ (where $p_D > 0$) yields

$$\int_\Omega \nabla p_D \cdot \nabla v_\phi\, dx = \int_\Omega p_D\, \nabla \log p_D \cdot \nabla v_\phi\, dx = \mathbb{E}_{x \sim p_D}\big[\nabla \log p_D(x) \cdot \nabla v_\phi(x)\big], \tag{13}$$

which gives equation 7.

**Boundary conditions.**  For Dirichlet conditions $v_\phi|_{\partial\Omega} = 0$, $\partial_n v_\phi$ is generally non-zero and the boundary flux term remains. For Neumann conditions $\partial_n v_\phi|_{\partial\Omega} = 0$, solvability requires $\int_\Omega s_\phi(x)\, dx = 0$, which is incompatible with nonnegative sources $s_\phi \geq 0$ unless $s_\phi \equiv 0$. This motivates the use of Dirichlet conditions or the corruption-driven proxy equation 7 in practice.

**Green representation.**  Under the same conditions, $v_\phi$ admits equation 5. In high-dimensional applications we approximate it via sample-based quadrature in representation space, using a regularized kernel $G_\varepsilon$ to avoid singularities at $x = y$.

## B  VISUALIZATION OF LATENT REPRESENTATIONS

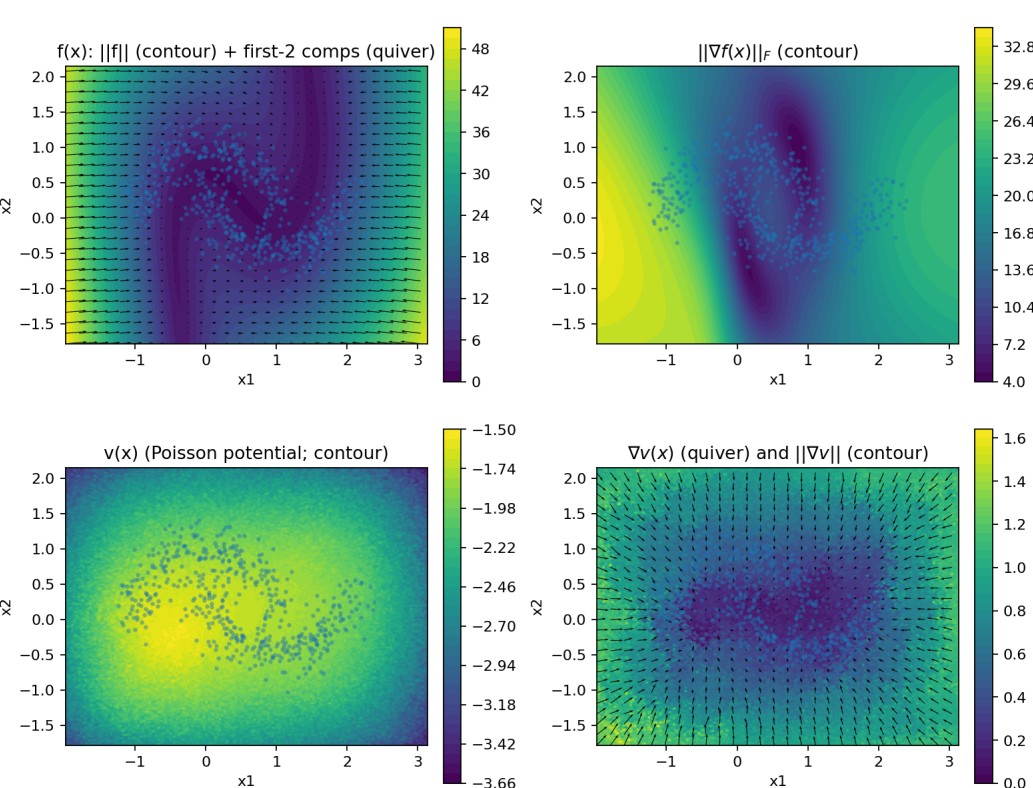

Figure 1: Two moons shape pattern with noise as corruption projection

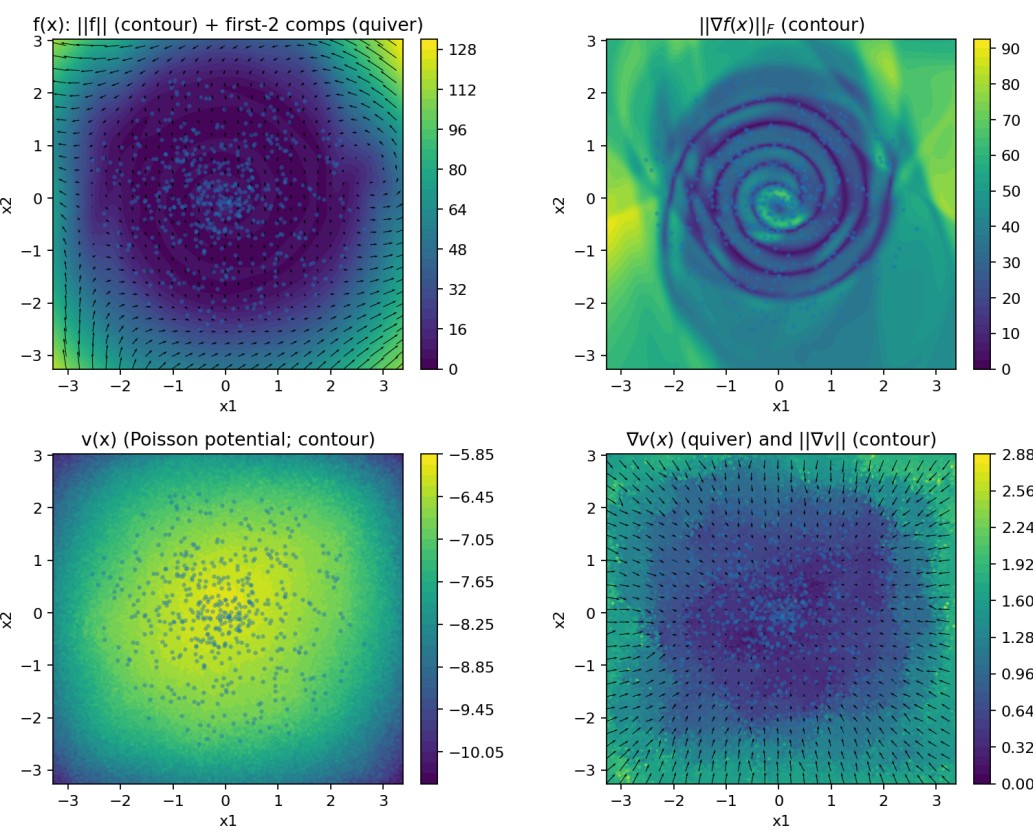

Figure 2: Spiral shape pattern with noise as corruption projection

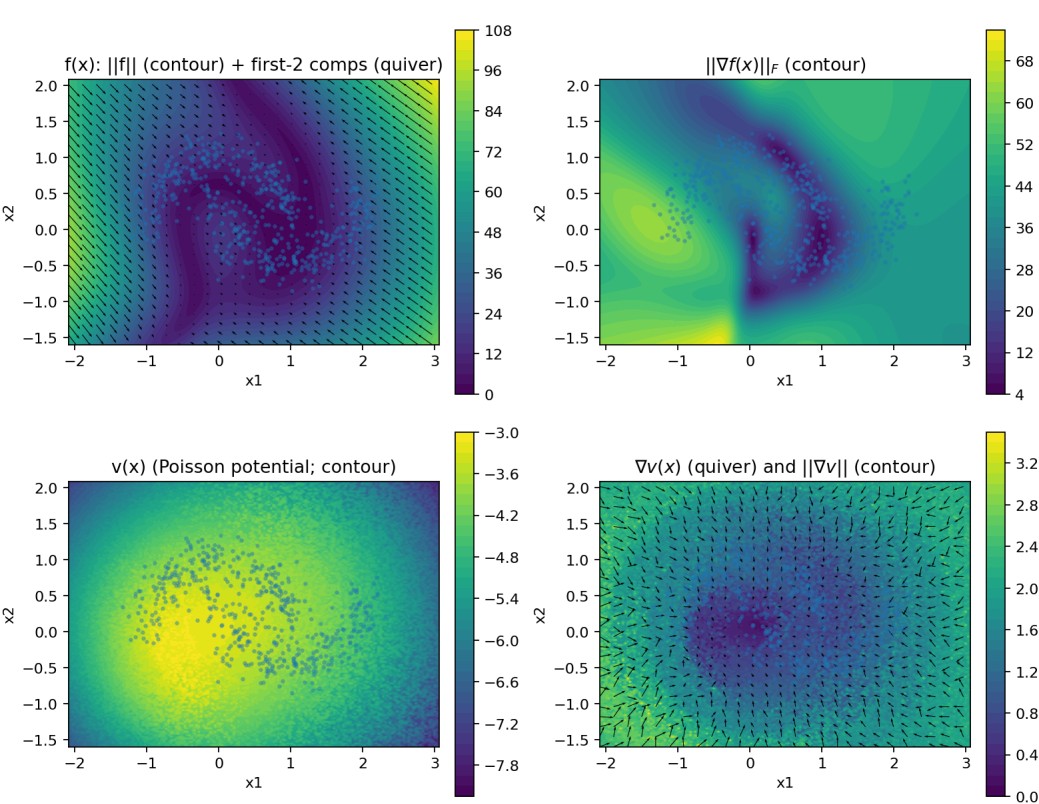

Figure 3: Two moon shape pattern with random rotation as corruption projection

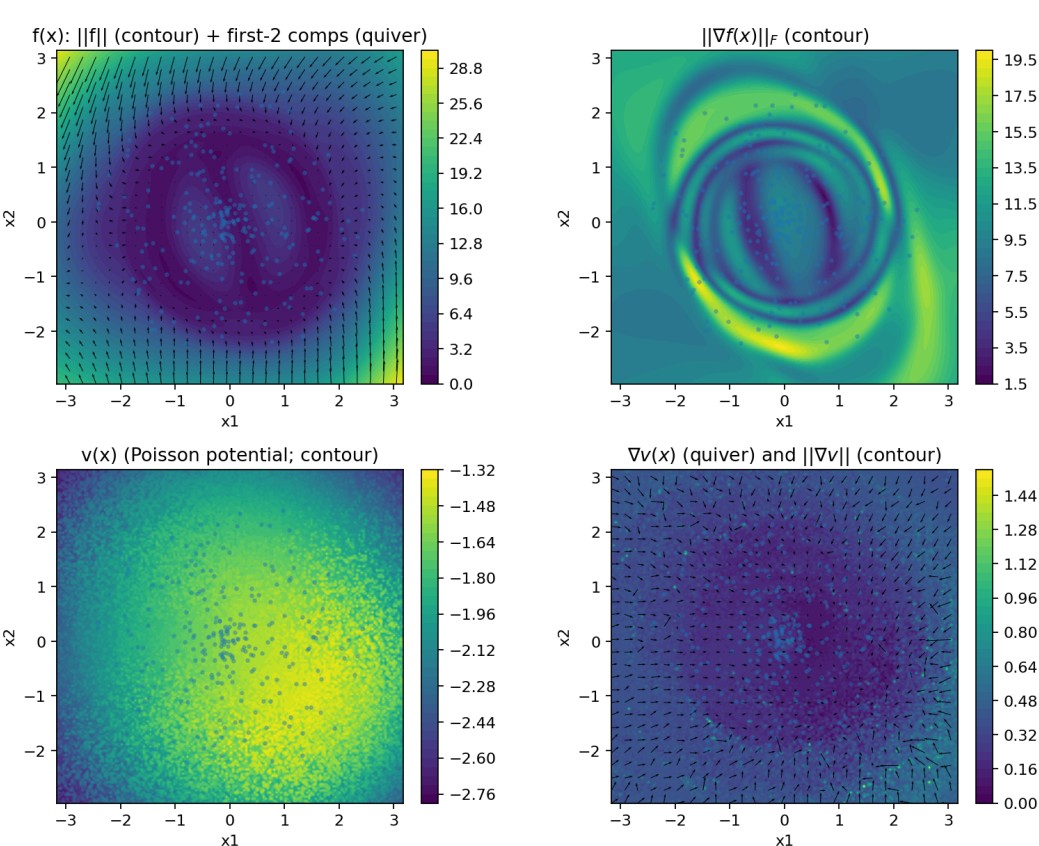

Figure 4: Spiral shape pattern with random rotation as corruption projection

