# OpenReview forum: "Poisson-Induced Potentials for Contractive representations"
_ICLR.cc/2026/Workshop/GRaM — ICLR 2026 Workshop GRaM Poster_

### Official Review · Reviewer_yxYT · 2026-02-09

**Rating:** 5
**Confidence:** 3

**Review:**

The authors study the regularisation term in Contractive Auto-Encoders (CAEs) and show that it can be reformulated via the solution of a Poisson equation. The resulting decomposition into a boundary-flux term and a score–potential coupling term is interesting. To make the formulation practical, the authors propose approximating the boundary-flux term using perturbation operators. However, the score-potential terms still depends on the unknown score of the distribution.

Overall, the paper is a reasonable fit for the workshop theme. That said, the interaction between the two terms in the decomposition and the geometry of different data distributions could be discussed in more depth.

I have a few comments:
- Currently, the authors focus a lot on the boundary-flux term and provide less discussion on the score-potential coupling. However, for distribution for which the density $p_D(x)$ decays to zero at the boundaries of the domain, the boundary-flux term becomes zero. In this setting, the regulariser is only driven by the score-potential coupling.
- The experiments are not really discussed in the main text (the Figures are not even linked). It would be good to provide a discussion and interpretation of the results in the Appendix. Also there are not experimental details.
- What is the loss function $\ell$ in Equation (2) ?
- The authors discuss the approximation of the boundary flux term. However, for practical examples the decomposition can not be computed as the score $\nabla \log p_D$ is generally unknown.


Minor points:
- The PDF states "Proceedings Track", but the paper is in the tiny paper track
- Abstract: I guess there is a word missing in the first sentence? "Contractive Auto-Encoders (CAE) orthogonal representations by penalizing the
Frobenius norm of the encoder Jacobian"
- Abstract: The grammar in the second and third sentence is wrong
- The citations are missing brackets.
- Line 85, page 2: You already introduced CAE in the previous section
- Line 98, page 2: I think it should be $\mathcal{R}_\text{CAE}$

**Pmlr Suitability:**

NA

---

### Official Review · Reviewer_iBsw · 2026-02-17
**Review of Poisson-Induced Potentials for Contractive representations**

**Rating:** 6
**Confidence:** 2

**Review:**

**Summary**: The paper reformulates the contractive auto-encoder (CAE) regularizer using a Poisson equation. This reformulation decomposes the regularizer into a boundary flux term and an interior score–potential coupling term. Based on this decomposition, the authors connect regularized auto-encoders to score-based modeling, such as diffusion models.

**Strengths**
- The decomposition of the CAE regularizer provides interesting insights that merit further exploration, especially the link between Jacobian-based regularization and learning on corrupted samples.

**Weaknesses**
- The authors approximate the Poisson potential via Monte Carlo estimation. I’m concerned about scalability, since the approach may require computing encoder Jacobians at many sampled points.
- The paper contains several grammatical errors and typos, which make it harder to follow.
- The experiments are limited to toy datasets (e.g., two moons and spiral).

**Pmlr Suitability:**

NA

---

### Official Review · Reviewer_ojNb · 2026-02-22
**Review of "Poisson-Induced Potentials for Contractive representations"**

**Rating:** 6
**Confidence:** 3

**Review:**

**Summary:**\
The paper presents a new interpretation of contractive autoencoders from the perspective of implicit generative modeling. Through a Poisson-based reformulation and the use of Green’s first identity, the contractive regularizer is decomposed into interior and boundary contributions. The resulting viewpoint emphasizes the role of out-of-distribution perturbations and negative samples in learning robust representations, connecting CAEs to denoising and diffusion-based approaches.

**Strengths:**
* The paper provides a meaningful theoretical bridge between contractive autoencoders and implicit generative modeling, connecting CAEs to denoising- and diffusion-based learning through a Poisson and PDE-based interpretation.
* The Poisson reformulation offers a geometric interpretation of the contractive penalty, decomposing it into interior and boundary effects and clarifying the role of out-of-distribution directions.

**Weaknesses&Questions:**
* While the high-level ideas are clear and insightful, the mathematical development and notation are at times difficult to follow. Streamlining the notation would significantly improve the overall readability of the paper.
* It remains unclear how the score-related term in the Poisson regularizer is handled in practice, and how it is combined with the proposed boundary-based approximation in the actual experiments.
* The use of corruption-induced displacements as a proxy for boundary normals is an interesting modeling choice, but its validity and limitations could be discussed more explicitly, especially when corruptions may not align with true normal directions of the data manifold.

**Pmlr Suitability:**

NA

---

### Meta-Review · Area_Chair_xAnf · 2026-02-26

**Decision:**

Accept

**Metareview:**

The reviewers highlight that the paper provides an interesting bridge between contrastive autoencoders and diffusion models vis the Poisson equation. The clarity and presentation issues are expected to be addressed by the authors and I am therefore happy to recommend acceptance.

**Relevance To Proceedings:**

Tiny paper — does not apply

**Relevance To Workshop:**

Yes — suitable for GRaM

---

### Decision · Program_Chairs · 2026-03-02

Accept (Poster)